# Effects of Multi-Pass Turning on Stress Corrosion Cracking of AISI 304 Austenitic Stainless Steel

**DOI:** 10.3390/mi13101745

**Published:** 2022-10-15

**Authors:** Yansong Zhang, Huan Xue, Yongchun Li, Xuelin Wang, Xinli Jiang, Chongwen Yang, Kewei Fang, Wenqian Zhang, Hui Jiang

**Affiliations:** 1Hubei Key Laboratory of Modern Manufacturing Quality Engineering, School of Mechanical Engineering, Hubei University of Technology, Wuhan 430068, China; 2School of Mechanical Science and Engineering, Huazhong University of Science and Technology, Wuhan 430076, China; 3Suzhou Nuclear Power Research Institute, Suzhou 215004, China; 4School of Mathematics and Statistics, Huazhong University of Science and Technology, Wuhan 430076, China

**Keywords:** multi-pass turning, austenitic stainless steel, roughness, micro-hardness, residual stress, stress corrosion cracking

## Abstract

Austenitic stainless steels are extensively used in mechanical engineering. The machined surface integrity has an essential influence on the stress corrosion cracking (SCC) performance of stainless steels. In this paper, the effects of multi-pass turning on the SCC susceptibility of AISI 304 austenitic stainless steel were investigated by correlating the SCC crack density to the machining-induced surface characteristics in terms of roughness, micro-hardness, and residual stress. In the multi-pass turning, the surface roughness and residual stress were the least after the double pass turning, and the surface micro-hardness was the maximum after the triple-pass turning. The SCC susceptibility was evaluated after SCC tests in boiling MgCl_2_ solution. The results showed that the weakest SCC sensitivity was observed in double-pass turning 304 stainless steel, while the most susceptible SCC was found in triple-pass turning. Compared with the double-pass turning, the increase in SCC sensitivity of triple-pass turning was attributed to the larger roughness, higher micro-hardness and greater residual tensile stresses.

## 1. Introduction

Austenitic stainless steels have been used in marine [1], petroleum [2], and nuclear power [3,4], owing to its excellent durability, high ductility, and great corrosion resistance [5,6]. However, the machining process could significantly affect the surface’s performance [7,8,9], such as micro-hardness [10], roughness [11], residual stress [12], and microstructure [13]. These machining-induced surface integrities can influence the service life and stress corrosion cracking (SCC) resistance of the components [14,15,16]. Therefore, the investigation of surface properties after machining is a very important issue.

Turning conditions have an impact on the microstructure and mechanical properties of the machined surface [17,18]. Zhang et al. [19] studied the effects of turning parameters on the residual stress, micro-hardness, and microstructure of AISI 304. Zhuang et al. [20] indicated that the residual stresses rise as the tool edge radius increases. Electroplusing [21], high-pressure jet [22], and liquid nitrogen [23]-assisted turning effectively reduce the surface roughness of AISI 304. In various nanofluid-assisted turning AISI 304 stainless steel, advantages were observed in terms of surface roughness, cutting forces, turning temperature, and flank wear [24].

Multi-pass machining is an excellent alternative solution for machining with large material removal rates and may be a feasible method to improve the integrity of the machined surface. Ding et al. [25] pointed out that the surface cracks of monocrystalline silicon were reduced and the machining accuracy was improved after multiple cutting. Hou et al. [26] reported that multi-step turning of Ti-6Al-4V titanium alloy was beneficial to reduce surface roughness. After multi-step cutting of a Ti-6Al-4V titanium alloy, the compressive residual stress increased and the residual stress distribution below the workpiece surface was improved [27]. More compressive residual stresses and more smooth machined surfaces were obtained in a multi-pass turning AISI D2 cold-work tool steel [28]. Moreover, multi-pass turning optimization models have been developed with the objectives of low energy consumption, high machining quality, and great efficiency [29,30,31]. However, the effects of multi-pass machining on the surface integrity of stainless steels have not been clearly studied.

The SCC of stainless steels is a major mode of material failure in engineered structures, which is influenced by surface integrity [1,2,3]. In recent years, many researchers have devoted themselves to the study of SCC in stainless steels [6,16]. Peng et al. [32] stated that the compressive residual stresses of AISI 304 austenitic stainless steel induced by low-temperature surface carburization inhibited the initiation of SCC. Laser-shock processing [33] and laser peening [34] resulted in compressive residual stresses and grain refinement on the machined surface of AISI 304 austenitic stainless steel, which effectively delayed the development of SCC. The dense dislocations inhibited the initiation of SCC in 304 stainless steel [35], while martensite reduced the resistance of AISI 304 stainless steel to SCC [36]. The relatively large roughness made SCC occur easily on the surface of stainless steel [37]. However, little work has been devoted to the effects of multi-pass turning on the SCC susceptibility of AISI 304 stainless steel.

This paper aimed to investigate the impact of multi-pass turning on the SCC resistance of AISI 304 stainless steel. The surface roughness, micro-hardness, and residual stresses were characterized after multi-pass turning. SCC tests were conducted in boiling MgCl_2_ to determine the SCC susceptibility of the machined surfaces. The current study correlated the mechanical properties produced by multi-pass turning with the SCC susceptibility.

## 2. Material and Method

### 2.1. Procedures

The flow chart of the experimental procedures and results analysis is shown in Figure 1. The starting conditions included workpiece material parameters, tool parameters, and turning condition. Firstly, multi-pass turning operations were performed on AISI 304 stainless steel. Then, physical performance measurements, SCC tests, and crack density measurements were carried out. Subsequently, the effects of multi-pass turning on roughness, residual stress, hardness, and crack density were analyzed. Finally, the influences of roughness, residual stress, and microhardness on the SCC susceptibility were investigated.

### 2.2. Material

The material used in this study was AISI 304 austenitic stainless steel, which was solution annealed at 1040 °C for 60 min and then cooled in air. The 304 stainless steel was held at 450 °C for 2 h, and then furnace cooled to eliminate the internal stress introduced during solution treatment. The surface microhardness of the unturned 304 stainless steel was measured to be about 190 HV. Table 1 presents the chemical composition of the 304 stainless steel (wt.%), which was measured by a direct-reading spectrometer (PDA7000, Shimadzu, Japan). The workpiece was a tubular form with inner and outer diameters of 39.0 mm and 60.5 mm, respectively.

### 2.3. Specimen Preparation

Turning operations were carried out on HTC2050 CNC (SMTCL, Shenyang, China). Grooves were machined on AISI 304 steel pipe with a 3 mm interval, with a turning speed of 600 m/min and a feed rate of 0.05 mm/r. The width and depth of the grooves were 2 mm and 5 mm respectively. Tool_1 (Sandvik, MGMN200 UE050, Sweden) was used to cut the grooves for separating and identifying the multi-pass turning areas. Tool_2 (Sandvik, TNMG160408-MM, Sweden) was used to turn the workpiece surface. The schematic diagram of the multi-pass turning surface is shown in Figure 2a. The workpiece after multi-pass turning and defined directions is presented in Figure 2b. The feed direction and turning direction are considered as axial direction and radial direction, respectively.

To keep the cutting edge sharp for each trail, a new Tool_2 cutting edge was applied to each pass of each specimen. With five turning groups, the turning parameters of fifteen specimens are given in Table 2. A double-pass turning depth of 0.1 mm meant that two turns were performed and that both the first and second turning depths were 0.1 mm. In the present study, to reduce heat accumulation during multi-pass turning, the workpiece was kept static for 30 min and allowed to air cool to room temperature after the previous turning pass. Then, the next turning pass began.

### 2.4. Examinations of Roughness, Residual Stress, and Micro-Hardness

After the turning procedure, the surface roughness in the axial direction was measured by an ultra-deep field microscope (Olympus, DSX 510, Tokyo, Japan). The sampling cutoff wavelength of the filter was taken as 250 μm in the measurement. The average roughness and standard deviation of 10 measurements on the microphotograph were obtained. Furthermore, the microscopic morphology of the machined surface was obtained.

The residual stresses on machined surfaces were measured by an X-ray stress diffractometer (Proto, iXRD, Windsor, ON, Canada), as illustrated in Figure 3. The parameters used in residual stress measurements are shown in Table 3. Mn-Kα was used as the X-ray radiation, the collimation type was a round collimation tube of 2 mm diameter, and the Bragg angle was 156.4°. The voltage and current applied for the measurements were 20 kV and 4 mA, respectively. Ψ angles were applied at 25°, 13.6°, 7.5°, 0°, −7.5°, −13.6°, and −25°, and the austenite diffraction peak was set to be {3 1 1}. For each specimen, the residual stress measurements were performed on the machined surface in the feed direction (axial direction) and turning direction (radial direction). The average residual stress value and standard deviation for each specimen were obtained from three residual stress measurements.

The micro-hardness experiments were conducted on a Vickers hardness tester (Qness, Q10A+, Golling an der Salzach, Austrian) as presented in Figure 4. The workpiece after multi-pass turning was divided into small samples by wire cutting. The surface hardness was measured directly on the small samples. The small sample was inlayed in resin and the cross section was smoothed with 1000 grit sandpaper before the cross-section hardness measurement. A testing force of 100 gf with a holding time of 10 s. Firstly, the micro-hardness of the machined surface was measured. Then, the micro-hardness tests were taken in the cross-section. The range of micro-hardness measurements in the subsurface was about 9 μm~200 μm below the machined surface. Since the indentation area occupied several micrometers, a micro-hardness within first 9 μm below the surface could not be tested. It should be emphasized that the distance between any two micro-hardness test points was at least five times the diagonal length of the Vickers indentation, which avoided any mutual influence among the tested points. During microhardness measurements, errors may be introduced by the experimental apparatus and data estimation. Three surface micro-hardness measurements values were averaged for surface micro-hardness, and the standard deviation was based on the three surface micro-hardness measurements. About 10 sets of microhardness data were measured for each specimen in cross sections within 9 μm~200 μm.

### 2.5. SCC Tests and Micro-Crack Measurements

The SCC tests were performed in a saturated MgCl_2_ solution boiling at 155.0 ± 1.0 °C, according to the standardized method ASTM G36-94 [38]. Approximately 800 mL of solution was prepared by adding 1200 g of MgCl_2_·6H_2_O to 30 mL of singly distilled water in a 2 L Erlenmeyer flask. The bottom of the flask was covered with a layer of glass beads to prevent bumping, and the flask was equipped with a thermometer to check the temperature of the solution. The MgCl_2_ solution was brought to a boil and maintained a temperature of 155.0 ± 1.0 °C. The specimens were exposed to the boiling solution without applying any external stress.

When the exposure time reached 30 min, the specimens were taken out, cleaned with anhydrous alcohol, and dried naturally. The cracks in the machined surface were measured under a microscope (KEYENCE, VK-X200K, Osaka, Japan) with magnification in the range of 200× to 1000×.

The density of cracks was calculated in the marked area of each sample. (1) Three regions of specimens soaked in boiling MgCl_2_ were examined for cracks. One picture was captured for each region. The image area was approximately 653 μm × 489 μm. (2) The length of each crack exceeding 20 μm was measured in each image, and the length of total cracks was obtained by adding individual crack length. (3) The crack density was calculated as the ratio of the length of total cracks to the measured area. (4) The average crack density of three images was obtained.

### 2.6. Analysis of Variance

Analysis of variance (ANOVA) is used to find the factors that have a significant influence on the data. In order to describe the degree of influence of turning speed, turning depth, and number of turning passes on surface roughness, microhardness, residual stress, and SCC, the experimental data were analyzed using ANOVA by MATLAB 2020a software. ANOVA operates based on sum of squares, F ratios, and *p* values. The sum of squares presents the deviation from the mean, and the F ratios indicate the relative importance of the factors. The *p* value is statistically significant within the 95% confidence interval (i.e., significance level α = 0.05). The factor is significant if the *p* value is lower than α. On the contrary, if the *p* value is higher than α, the factor is unimportant. The turning passes and the total turning depth were correlated. Therefore, the significance of the interaction between turning depth and turning passes was also analyzed by ANOVA.

## 3. Results

### 3.1. Roughness

Multi-pass turning has a remarkable impact on surface roughness. Figure 5 presents the results of average surface roughness and standard deviation for different turning conditions. Among the tests, the maximum surface roughness appeared in single-pass turning with 2.4 µm, and the minimum surface roughness occurred in double-pass turning with 0.7 µm, as shown in Figure 5. The average roughness values after single, double, and triple-pass turning were 2.26 μm, 0.74 μm, and 1.18 μm, respectively. It could be observed that the roughness after single-pass turning was the largest. The surface roughness after double-pass turning was the minimum, while the surface roughness after triple-pass turning was slightly higher than that after double-pass turning. Comparing double-pass turning of T6 with single-pass turning of T3, triple-pass turning of T11 with double-pass turning of T7, the roughness of double-pass turning was excellent at the same turning speed and total depth of cut.

The microscopic images of the surface after multi-pass turning are presented in Figure 6. As shown in Figure 6a,b chip tumors were observed on the surface of single-pass turning, which led to rough machined surfaces. Figure 6c–f show the machined surfaces of double and triple-pass turning. Scratches left by the turning and fine chip grains could be clearly observed.

### 3.2. Residual Stress

The surface residual stresses and standard deviation values in the axial and radial direction under multi-pass turning conditions were obtained, as listed in Figure 7. In the axial direction, the residual stresses were compressive after single- and double-pass turning, while in the radial direction, tensile residual stresses were found in each pass turning.

In both the axial and radial directions, the residual stresses were a minimum after double-pass turning, while the surface residual stresses were a maximum after triple-pass turning. Compared with single-pass turning, the surface residual stresses in the axial and radial directions decreased by 69 MPa and 157 MPa on average for the double-pass turning. The residual stresses in triple-pass turning increased by an average of 202 MPa in the axial direction and an average of 78 MPa in the radial direction compared with single-pass turning. Comparing double-pass turning of T6 with single-pass turning of T3, triple-pass turning of T11 with double-pass turning of T7, the residual stress of double-pass turning was relatively small at the same turning speed and total cutting depth.

### 3.3. Micro-Hardness

The turning process may cause changes in microhardness due to plastic deformation and thermal effects. The surface microhardness and standard deviation after multi-pass turning are illustrated in Figure 8. The surface microhardness increased and then decreased with the turning speed for double and triple-pass turning, while the surface microhardness after single-pass turning presented an increase with the increase in speed. The surface microhardness after single-, double-, and triple-pass turning decreased and then went up with the turning depth. At the same turning parameters, the surface microhardness rose as the number of turning passes increased in single-, double-, and triple-pass turning.

According to the micro-hardness measurement results, the micro-hardness was obtained at different depths under multi-pass turning conditions, as shown in Figure 9. Deformation caused by hard turning could affect the hardness of the surface and sub-surface layers. From Figure 9a–e, the surface micro-hardness increased by 6.9%, 2.3%, 0.9%, 3.6%, and 2.1% after double-pass turning compared with single-pass turning, correspondingly, and the surface micro-hardness improved by 4.0%, 3.7%, 4.5%, 4.6% and 2.5% after triple-pass turning compared with double-pass turning, respectively.

The average micro-hardness of the surface after single-pass turning was 419 HV, while the average micro-hardness after double and triple-pass turning were 433 HV and 449 HV, respectively. As the number of turning passes increased, the surface micro-hardness tended to rise, while the micro-hardness of the subsurface layers did not differ significantly. Multi-pass turning had essentially no effect on the depth of the hardened layer, and the penetration depth of the turning hardening was about 160 µm. The micro-hardness decreased sharply within the 0~20 µm below the machined surface, while beyond 20 µm, the micro-hardness decreased slowly to approximately 190 HV. Measurements and readings may cause errors in the microhardness. However, the errors did not change the trend of microhardness below the surface.

### 3.4. Micro-Crack

The density of cracks, which was used to evaluate the SCC initiation susceptibility of the machined surface, was sensitive to the turning parameters and turning passes. Table 4 demonstrates the variations of cracks density in multi-pass turning. The crack density of T13 in triple-pass turning was the largest, which was 5987 μm/mm^2^. The average crack densities of the single-pass turning and triple-pass turning were 525 μm/mm^2^ and 2622 μm/mm^2^, respectively. No crack was observed in single-pass turning of T1 and T5, and the crack was not detected on the machined surface after the double-pass turning. The average crack densities of single-pass turning were slightly larger than that of double-pass turning, while the average crack densities of triple-pass turning were far greater than that of double-pass turning and single-pass turning.

The cracks of the specimens immersed in boiling MgCl_2_ for 30 min are illustrated in Figure 10. The cracks were mainly perpendicular to the radial direction due to the high tensile stress in the radial direction. Fine cracks and micro-pits were distributed on the surface, as shown in Figure 10a. The appearance of micro-pits was the result of machining defects. Figure 10b–d demonstrate that no cracks were observed on the turned surface. Turning scratches and machining defects were detected, as shown in Figure 10c,d. As presented in Figure 10e,f, more cracks were found on the surfaces of triple-pass turning. Cracks were generated in part of single-pass turning samples, no crack initiated in double-pass turning, and a great number of cracks were produced in triple-pass turning.

## 4. Discussion

### 4.1. Machining-Induced Roughness

Surface roughness is one of the main consequences of the turning process and plays an important role in stress corrosion crack initiation. Multi-pass turning reduced surface roughness and the ANOVA between the number of passes and surface roughness showed a *p* value below 0.05 (Table 5). The factor of h*pass presented the interaction of turning depth and turning pass. A *p* value less than 0.05 indicates that the turning pass had an impact on surface roughness. The *p* values of turning speed, turning depth, and the interaction of turning depth and turning pass were greater than 0.05, which suggested that turning speed, turning depth, and the interaction had little effect on surface roughness.

It should be pointed out that for the same total depth of turning, which is relating the single turning pass of T3 with the double-turning pass of T6, and the triple-turning pass of T11 with the double-turning pass of T7, the roughness of the double-turning pass was always lower. Turning passes and turning depth were correlated. Further experiments are needed to study the effect of different turning passes on roughness under the same turning speed and total turning depth.

As shown in Figure 6, the chip tumors attached to the machined surfaces after single-pass turning resulted in greater roughness. The number of chip tumors on the surface of double and triple-pass turning decreased. However, the roughness after triple-pass turning was greater than that after double-pass turning. The hardening caused by the double-pass turning led to deeper scratches after the triple-pass turning, which may cause the roughness to be larger after the third pass turning. Changes of microstructure and properties of near-surface layers in previous pass turning may have influenced the surface roughness of latter pass turning.

### 4.2. Effect of Residual Stress

The ANOVAs for the effects of the factors on the axial and radial residual stresses are given in Table 6 and Table 7, respectively. From the *p* value, the effect of turning passes on residual stresses was remarkable, in both the axial and radial directions, but the effect of the interaction of turning depth and turning pass on residual stresses was not important. Turning speed had no obvious effect on axial residual stresses, but it had a distinct impact on radial residual stresses. This was could be attributed to the turning force in the radial direction being larger than in the axial direction [19]. The different mechanical loads might explain the different residual stress values in the axial and radial directions.

The influences of turning on residual stress are complex. The residual stresses generated by the turning process are the result of a combination of several factors, including but not limited to turning parameters, the properties of the material, tool wear, and lubrication conditions [17]. Increasing the hardness of the workpiece converted the tensile residual stresses on the surface into compressive residual stresses and generated more compressive residual stresses on the subsurface of the workpiece [39]. However, Capello [40] showed that the tensile residual stress tended to rise with increasing hardness of the workpiece. When the hardness of the workpiece increased, the intense friction between the back tool face and the workpiece led to greater plastic deformation and more heat.

Plastic deformation could account for generating compressive residual stresses, while a thermal effect could produce tensile residual stresses. The thermal effect and plastic deformation resulting from the turning process affect the final state of the residual stresses. The hardness of the surface layer of the workpiece increased after the previous pass turning, which caused an increase in plastic deformation during the next pass turning. The decrease of residual stress was attributed to more severe plastic deformation in double-pass turning. However, after the previous turning, the microhardness of the surface layer increased, which might lead to more intense extrusion and friction between the tool and the workpiece, resulting in more heat. Therefore, the thermal effect may dominate the increase of tensile residual stress in triple-pass turning.

### 4.3. Micro-Hardness Analysis

An increase in micro-hardness was observed in the near-surface area of the turning samples, regardless of the turning parameters and turning passes. Severe plastic deformation could cause hardening of the machined surface of the specimen. Grain refinement, formation of twins, and increase in dislocation density may contribute to surface hardening. Compared with the unturned 304 austenitic stainless steel, the average surface micro-hardness increased by 120.5%, 127.9%, and 136.3% after single-, double-, and triple-pass turning, respectively. As listed in Table 8, the *p* values of turning speed, turning depth, and turning pass were less than 0.05, with values of 0.0036, 0.0001, and 0.0002, respectively. Turning speed, turning depth, and turning pass had a significant effect on the hardness of the turned surface. The *p* value for the interaction of turning depth and turning pass was 0.1095, which suggested that the interaction of turning depth and turning had no significant effect on microhardness.

As shown in Figure 8, when the turning speed and turning depth were the same, the surface microhardness after triple-pass turning was higher than that after single- and double-pass turning. However, under the same turning speed and turning depth, multi-pass turning had no significant effect on the micro-hardness of the subsurface layer, as shown in Figure 9. After the previous pass turning, the superficial surface layer of the workpiece hardened, which led to a greater surface micro-hardness after the subsequent turning pass. In the single-, double-, and triple-pass turning, the increase in micro-hardness with increasing turning pass was a result of the accumulation of turning hardening, which resulted in a harder surface micro-hardness after the latter pass turning than previous pass turning.

### 4.4. Correlation between Surface Properties and SCC in Multi-Pass Turning

The importance of turning speed, turning depth, and turning pass on crack density was evaluated by ANOVA (Table 9). The *p* value for turning speed was 0.7, while the *p* values for turning depth, turning pass, and the interaction of the turning depth and turning pass were 0.0345, 0.0015, and 0.0228, respectively. The ANOVA results showed that turning speed did not play a significant role in crack density.

Multi-pass turning changed the properties of the machined surface such as roughness, micro-hardness, and residual stress, which were direct factors affecting the stress corrosion resistance. The SCC experiment revealed the cracking susceptibility of the 304 stainless steel in an MgCl_2_ medium. The contact area was larger where the roughness was greater, which collected more chloride. In areas of indentation caused by processing, grain refinement might occur by nano crystallization, which could cause more susceptibility to cracking [41]. Yang et al. [42] stated that the sensitivity to SCC was significantly increased by the increase in surface hardness. The hardened layer caused by the turning process promoted the development of cracks and reduced the stress corrosion resistance of 304 stainless steel. The residual tensile stress of machined surface in the radial direction was much higher than that in the axial direction. Correspondingly, the cracking direction was mainly vertical to the machining striations as shown in Figure 10. It indicated that the initiation of microcracks was associated with the level of residual tensile stresses [16].

As given in Table 4, with the increase of turning passes from single-pass to triple pass, the crack density decreased and then increased. Turning experiments and surface properties testing process may cause errors in the results due to instrumentation factors. The following speculations were based on the experimental measurements in this paper. The large surface roughness, radial tensile residual stresses, and hardened layer may facilitate crack expansion in single-pass turning, while residual compressive stresses in the axial direction enhanced stress corrosion resistance. The compressive residual stress dominated the inhibition of crack expansion, which resulted in a relatively low density of cracks in single-pass turning conditions. Compared with single-pass turning, double-pass turning increased surface micro-hardness and reduced roughness and residual stresses. The lower residual stresses and roughness may result in no cracks on the machined surface in double-pass turning. The tensile residual stress in the axial and radial directions and micro-hardness weakened the stress corrosion resistance of 304 stainless steel in third pass turning, which could cause the density of cracks after triple-pass turning was much larger than that after single-pass and double-pass turning.

The Pearson correlation coefficient is a mathematical analysis that describes the strength of the relationship between two variables. The Pearson correlation coefficient r takes values in the range (−1, 1). An r less than 0 indicates that the two variables are negatively correlated, while an r greater than 0 means that the two variables are positively correlated. The closer the absolute value of r is to 1, the stronger the correlation between the two variables. N represents the number of data points each analysis. Pearson correlation coefficients were calculated for roughness, surface hardness, axial residual stress, and radial residual stress with crack density, respectively, as listed in Table 10. The correlation of roughness and crack density were negative and extremely weak, while surface microhardness and crack density showed a positive and moderate correlation. Both correlations of axial residual stress and radial residual stress with crack density were positively and strongly correlated.

A previous investigation [41] suggested a positive correlation between roughness and crack density, while the Pearson correlation coefficient showed that they were negatively correlated, which was due to the dominant role of residual stresses in initiating SCC. Zhang et al. [43] also stated a strong correlation between residual stress and crack density. In this work, compared with single-pass turning, the roughness decreased and residual stress increased after triple-pass turning. However, the change in micro-hardness was not obvious. Therefore, residual stress was the main factor affecting the SCC susceptibility. That was the reason that the crack density increased when the roughness was reduced.

Obviously, the state of biaxial stresses caused by machining had a considerable effect on the initiation of SCC. Figure 11 shows the relationship between SCC initiation and biaxial residual stress distribution. The solid black squares signified that cracks were generated, while the red circles meant that no crack was initiated. From the figure, the area of SCC initiation was concentrated in the upper right of the coordinate system, while the area without cracks was concentrated in the lower left of the coordinate system. Cracks were found at the locations with larger residual stresses in both the axial and radial directions, indicating high SCC susceptibility in the region. Similar results were reported in a study of 316 stainless steel [44]. Rhouma et al. [45] indicated that a threshold straight line divides the biaxial stress coordinate system into a crack-presence region and a crack-free region. The threshold, which was less than 270 MPa, was the sum of the axial and radial stresses. Similar results were obtained in the current paper, and the threshold value was about 240 MPa for 304 austenitic stainless steel.

Meanwhile, the trend of crack density with the combined residual stresses is illustrated in Figure 12. When the combined residual stress was relatively small, no cracks were generated on the surface. As the combined residual stress increased to about 320 MPa, the surface crack density sharply grew with the residual stress. Therefore, it could be surmised that the larger residual tensile stress on the machined surface increased the SCC susceptibility of 304 stainless steel in the high-temperature MgCl_2_ solution.

## 5. Conclusions

In this paper, the effects of single-, double-, and triple-pass turning of 304 stainless steel on stress corrosion resistance were investigated. It is difficult to state the nonlinear trend with three turning passes, and more turning passes are needed in future studies. According to the single-, double-, and triple-pass turning, the main conclusions are listed as follows:

Compared with single-pass turning, the machined surfaces were smoother after double-pass turning, and the surface microhardness was higher after double- and triple-pass turning. The residual stresses in both axial and radial directions were lowest after double-pass turning compared with single- and triple-pass turning.

The increase of surface roughness, micro-hardness, and residual stress contributed to greater SCC susceptibility. The influence of surface roughness on the initiation of SCC was relatively weak, while the axial and radial residual stresses played a more pronounced role.

The relatively minor residual stresses and roughness were observed on the surfaces of double-pass turning, which reduced the SCC susceptibility to the aggressive Cl^−^ environment. The SCC of the machined surface after triple-pass turning was the most susceptible, which could be explained by the fact that the triple-pass turning led to higher roughness, more hardening of the surface, and greater residual tensile stresses.

## Figures and Tables

**Figure 1 micromachines-13-01745-f001:**
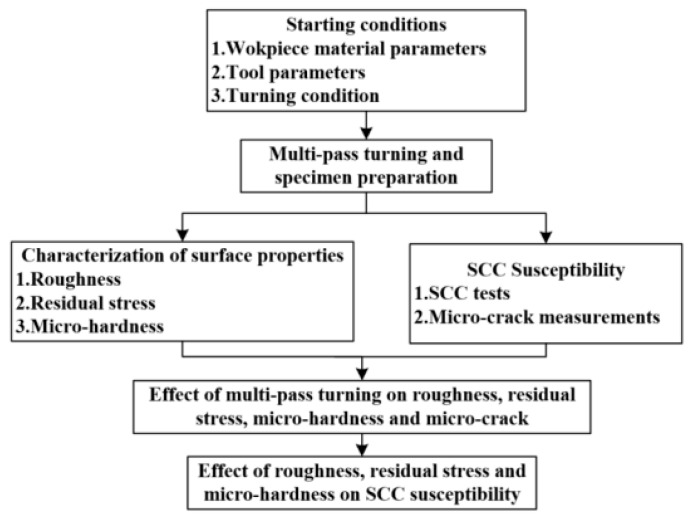
The flow chart of experiments and analyses.

**Figure 2 micromachines-13-01745-f002:**
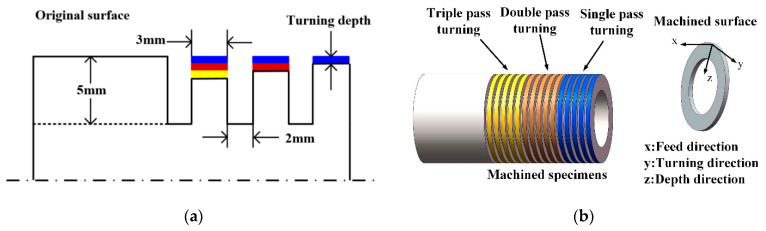
Schematic diagram showing multi-pass turning for the specimens: (**a**) multi-pass turning process and (**b**) workpiece after multi-pass turning and directions.

**Figure 3 micromachines-13-01745-f003:**
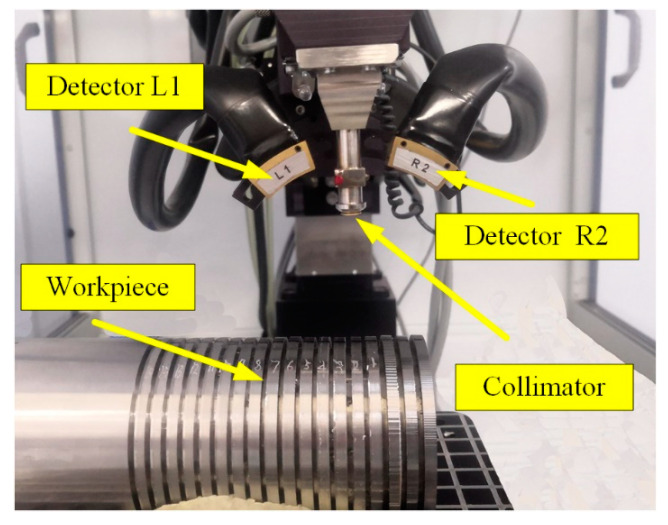
Residual stress measurement process.

**Figure 4 micromachines-13-01745-f004:**
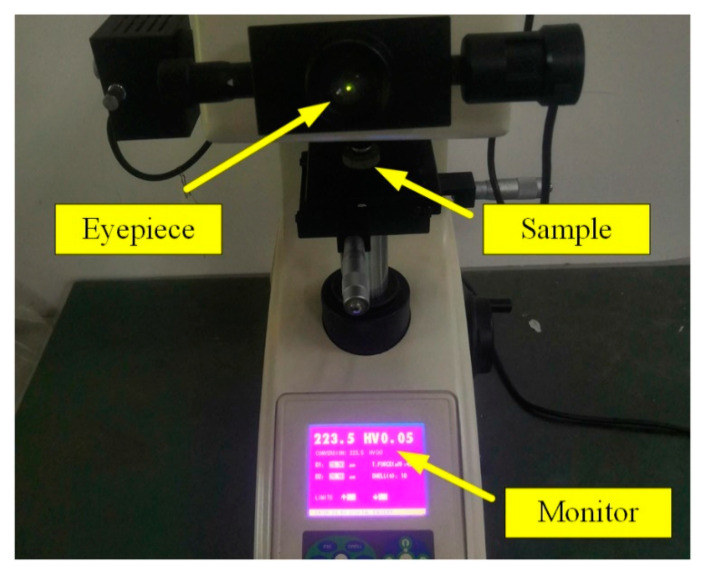
Micro-hardness measurement process.

**Figure 5 micromachines-13-01745-f005:**
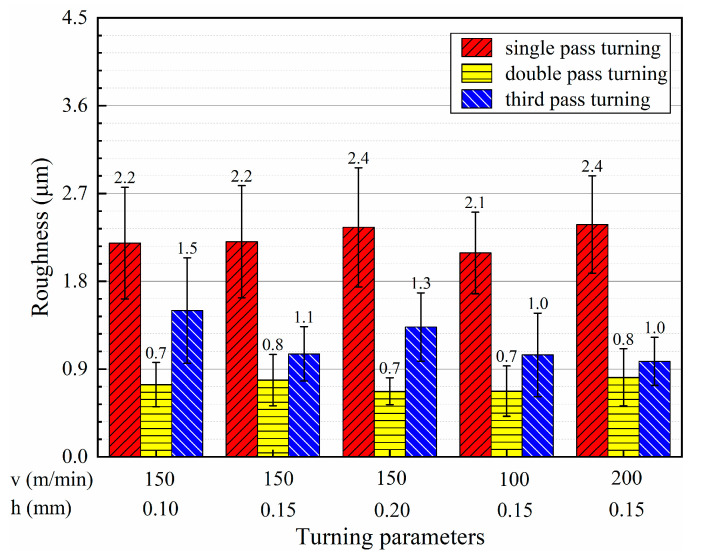
Average roughness of the machined surfaces under different turning conditions.

**Figure 6 micromachines-13-01745-f006:**
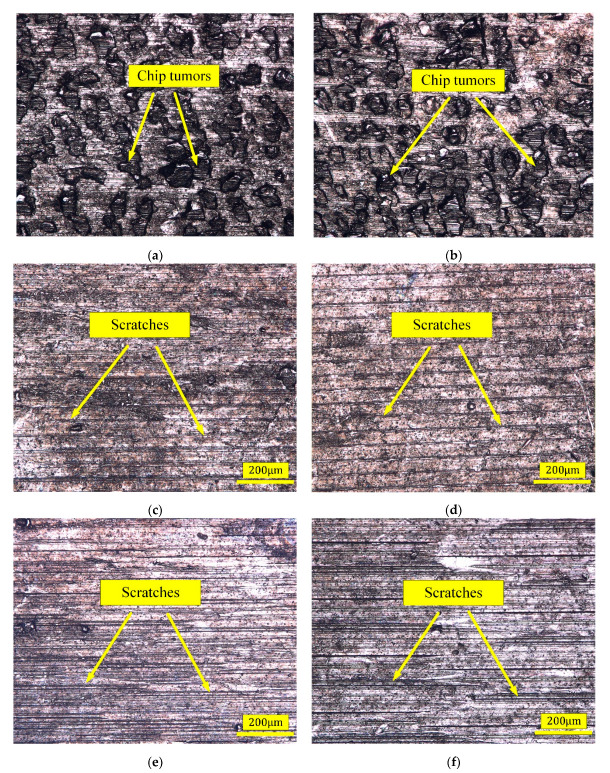
Microscopic morphology of the machined surface after multi-pass turning: (**a**) single-pass turning of T1, (**b**) single-pass turning of T2, (**c**) double-pass turning of T6, (**d**) double-pass turning of T7, (**e**) triple-pass turning of T11, and (**f**) triple-pass turning of T12.

**Figure 7 micromachines-13-01745-f007:**
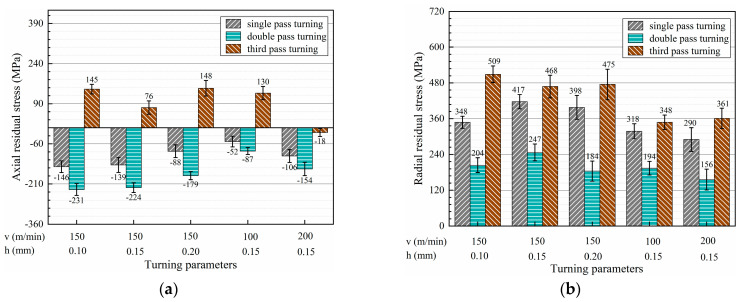
Surface residual stresses under multi-pass turning conditions: (**a**) surface residual stress in the axial direction and (**b**) surface residual stress in the radial direction.

**Figure 8 micromachines-13-01745-f008:**
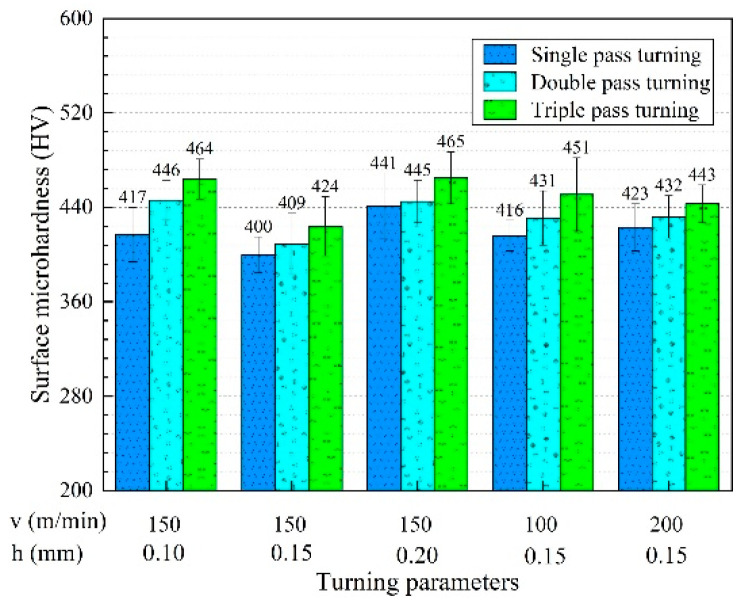
Surface microhardness under multi-pass turning conditions.

**Figure 9 micromachines-13-01745-f009:**
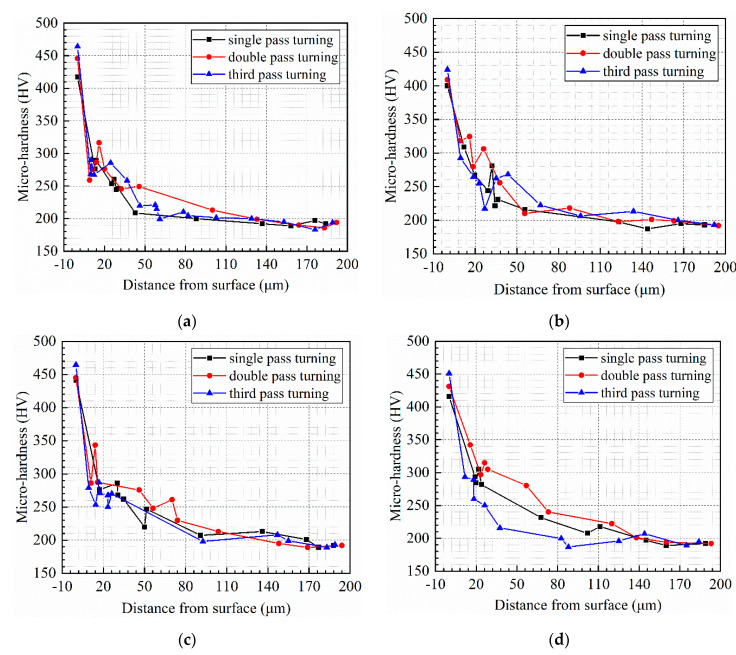
Profile of micro-hardness distribution under multi-pass turning conditions: (**a**) turning No. of T1, T6, and T11, (**b**) turning No. of T2, T7, and T12, (**c**) turning No. of T3, T8, and T13, (**d**) turning No. of T4, T9, and T14, (**e**) turning No. of T5, T10, and T15.

**Figure 10 micromachines-13-01745-f010:**
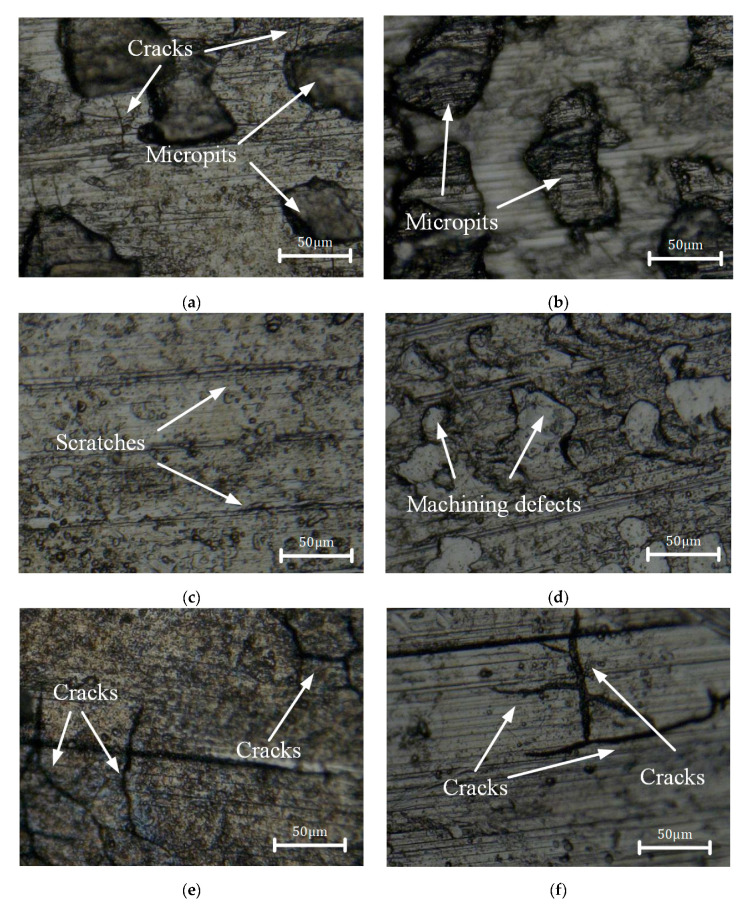
Cracks of the machined surface after multi-pass turning: (**a**) single-pass turning of T3, (**b**) single-pass turning of T5, (**c**) double-pass turning of T8, (**d**) double-pass turning of T10, (**e**) triple-pass turning of T13, (**f**) triple-pass turning of T15.

**Figure 11 micromachines-13-01745-f011:**
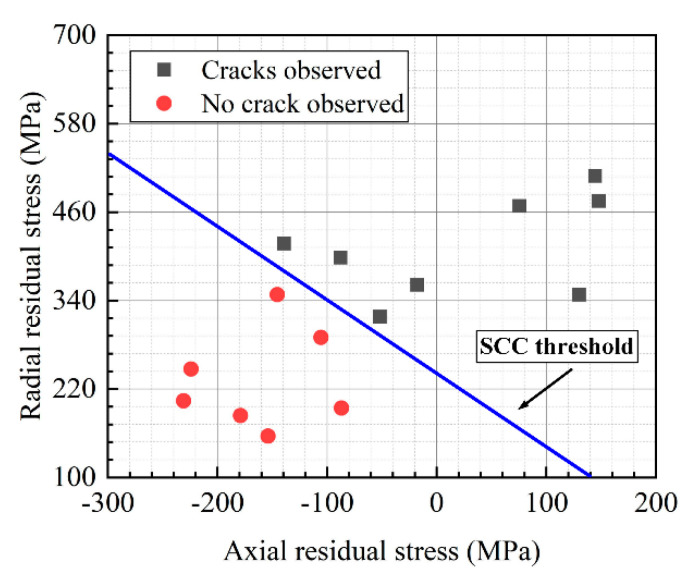
Relationship between SCC initiation and biaxial residual stresses.

**Figure 12 micromachines-13-01745-f012:**
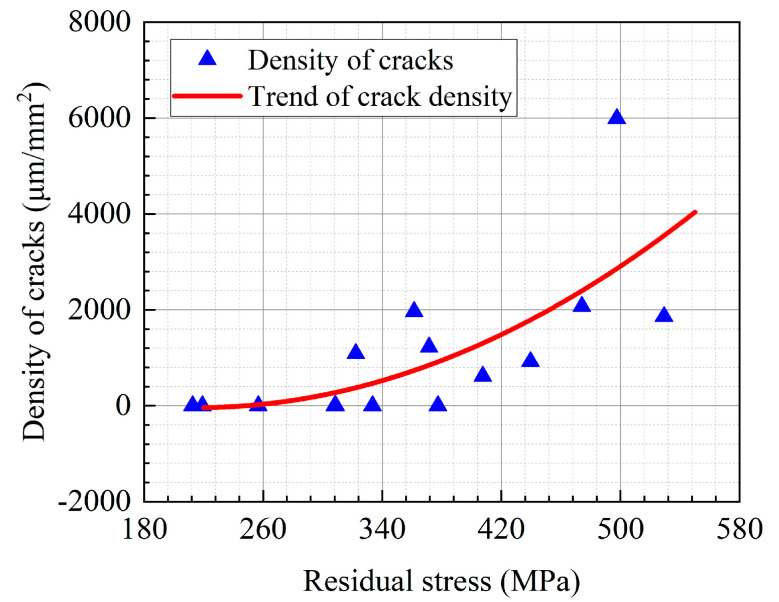
Correlation between crack density and residual stress in AISI 304 stainless steel.

**Table 1 micromachines-13-01745-t001:** Chemical composition of the AISI 304 stainless steel (wt.%).

C	Si	P	Mo	Mn	Cr	Ni	Fe
0.0389	0.446	0.0225	0.172	0.675	18.3	8.53	Balance

**Table 2 micromachines-13-01745-t002:** Turning parameters for specimen preparation.

Turning No.	Turning Speed v (m/min)	Turning Depth h (mm)		
		Single-Pass Turning	Double-Pass Turning	Triple-Pass Turning
T1	150	0.10	-	-
T2	150	0.15	-	-
T3	150	0.20	-	-
T4	100	0.15	-	-
T5	200	0.15	-	-
T6	150	0.10	0.10	-
T7	150	0.15	0.15	-
T8	150	0.20	0.20	-
T9	100	0.15	0.15	-
T10	200	0.15	0.15	-
T11	150	0.10	0.10	0.10
T12	150	0.15	0.15	0.15
T13	150	0.20	0.20	0.20
T14	100	0.15	0.15	0.15
T15	200	0.15	0.15	0.15

**Table 3 micromachines-13-01745-t003:** The parameters used in residual stress measurements.

Parameter	Value
Radiation	Mn-Kα
Collimator type	Round, 2 mm diameter
X-ray tube voltage/current	20 kV/4 mA
Ψ angles	25°, 13.6°, 7.5°, 0°, −7.5°, −13.6°, −25°
Bragg angle (2θ)	156.4°
diffraction peak	{3 1 1}

**Table 4 micromachines-13-01745-t004:** The observed crack density of multi-pass turning.

Turning Speed (m/min)	Turning Depth (mm)	Density of Cracks (μm/mm^2^)		
		Single-Pass Turning	Double-Pass Turning	Triple-Pass Turning
150	0.10	No crack observed	No crack observed	1861
150	0.15	921		2072
150	0.20	617		5987
100	0.15	1090		1223
200	0.15	No crack observed		1965

**Table 5 micromachines-13-01745-t005:** ANOVA table for surface roughness (Ra).

Factors	Sum of Squares	Degrees of Freedom	Mean Square	F Ratio	*p* Value	Remark
v	0.0289	2	0.0144	1.8571	0.2689	Not significant
h	0.02	2	0.01	1.2875	0.3705	Not significant
pass	4.7937	2	2.3968	308.16	0.00004	Significant
h*pass	0.1582	4	0.0396	5.0857	0.0721	Not significant
Error	0.0311	4	0.00078			
Total	6.3893	14				

**Table 6 micromachines-13-01745-t006:** ANOVA table for axial residual stress on the surfaces.

Factors	Sum of Squares	Degrees of Freedom	Mean Square	F Ratio	*p* Value	Remark
v	16,636.2	2	8318.1	4.25	0.1024	Not significant
h	4890.9	2	2445.4	1.25	0.3787	Not significant
pass	196,530.4	2	98,265.2	50.22	0.0015	Significant
h*pass	12,632.5	4	3158.1	1.61	0.3271	Not significant
Error	7826.4	4	1956.6			
Total	238,071.3	14				

**Table 7 micromachines-13-01745-t007:** ANOVA table for the radial residual stress on the surfaces.

Factors	Sum of Squares	Degrees of Freedom	Mean Square	F Ratio	*p* Value	Remark
v	20,268.7	2	10,134.3	27.27	0.0047	Significant
h	1186.9	2	593.4	1.6	0.3092	Not significant
pass	136,720.4	2	68,360.2	183.93	0.0001	Significant
h*pass	8803.2	4	2200.8	5.92	0.0566	Not significant
Error	1486.7	4	371.7			
Total	180,436.4	14				

**Table 8 micromachines-13-01745-t008:** ANOVA table for surface micro-hardness.

Factors	Sum of Squares	Degrees of Freedom	Mean Square	F Ratio	*p* Value	Remark
v	938.89	2	469.44	30.4	0.0038	Significant
h	2592.89	2	1296.44	83.94	0.0005	Significant
pass	2036.32	2	1018.16	65.92	0.0009	Significant
h*pass	244.09	4	61.02	3.95	0.1059	Not significant
Error	61.78	4	15.44			
Total	5172.4	14				

**Table 9 micromachines-13-01745-t009:** ANOVA table for density of cracks.

Factors	Sum of Squares	Degrees of Freedom	Mean Square	F Ratio	*p* Value	Remark
v	182,254.2	2	91,127.1	0.39	0.7	Not significant
h	4,090,754.9	2	2,045,377.4	8.76	0.0345	Significant
pass	23,280,775.0	2	11,640,387.5	49.87	0.0015	Significant
h*pass	9,455,450.4	4	2,363,862.6	10.13	0.0228	Significant
Error	933,651.1	4	233,412.8			
Total	34,866,544.9	14				

**Table 10 micromachines-13-01745-t010:** The results of Pearson correlation coefficient.

Surface Properties	N	r	Correlated	Remark
Roughness	15	−0.0059	Negatively	Extremely weak
Micro-hardness	15	0.5080	Positively	Moderate
Axial residual stress	15	0.7526	Positively	Strong
Radial residual stress	15	0.6832	Positively	Strong

## Data Availability

Not applicable.

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
