# Peer review of "Effects of Multi-Pass Turning on Stress Corrosion Cracking of AISI 304 Austenitic Stainless Steel"

_micromachines, 2022, doi:10.3390/mi13101745_

Round 1

Reviewer 1 Report

Generally good paper.  See attached for comments and suggestions for improvement.

Reviewer 2 Report

Overall: This could be a great paper but is written like a report: A) Relationship between Roughness, Residual stress, and Hardness is not defined.  Common Parameters effecting interaction must clearly be defines. B) Add Flow Chart of the entire system including Single, double, and triple pass and explain why double pass is best and recommended. C) Table 2 is not sufficient.  D)  Reorganize the paper Move Residual stress (line 179)  after Roughness (Line 159), Add Hardness after Residual stress . E) Paper Goal ( Line 67-71) does not support the Abstract and conclusion. Both Conclusion and Abstract must be re-written. 

DETAILS:

Note 1 :Introduction (Line 24) is good, however needs to describe the more details such as  relationship between roughness, and residual stress as surface traction (is not explained)

Note 2 : Figure 1- Please develop a new table and describe all parameters .

Note 3. I suggest plot of Temperature -time -History (TTH) by measurement of temperature or analysis be plotted as new Figure showing Double Pass Turning compared. 

Note 4 : Figure 3- Need More Detail- Please describe why Double pass has less roughness. What causes it. Explanations are not clear in Line 155-157 “ The surface roughness  after double pass turning was the minimum, while the surface roughness after triple pass  turning was slightly higher than that after double pass turning.  You May use   

Note 4 : Figure 4. Shows no difference in Micro hardness between Single, double, and triple .  Why? Please explain. If Residual stress has any effect

Note 5: Figure  5A why Residual stress is negative in single and third pass.  In axial direction.  Please explain is not clear what caused it. Figure  5B Why Radial stress in double pass is reduced.  This needs to be clearly explained as what parameters are casing this behavior.

Note 6 : Table 4.  Explain the behavior. Of Double pass Crack density. It simply does not make sense.).

Note 7 : Figure 6 EXPAND IT.  As the explanation of double pass 2 . May be in Double pass crack is lower due to grain boundary (Intergranular)  vs. Single and Triple Pass Grain crystallization (Trans granular crack)

Note 8 : Table 6-8.  ANOVA: Describe ANOVA technique in detail, Why Pearson correlation coefficient used

Note 9. Plot all the important parameters causing) Rouhness, b) residual stress, and c) Hardness. Plot sensitivity of all parmaters in Table 6-9 in the form of CDF (Cummulative Distribution Function )  and in more detail Plot 

Round 2

Reviewer 1 Report

I think the manuscript is now minor revisions. The technical work is acceptable, the analysis is weak in places, but acceptable. The conclusions and inferences are not always justified, and this is where I had problems. I disagree with the authors in many places, but that debate will be up to the general readership to engage in.

Reviewer 2 Report

Overall: This Paper describes the effect of machining depth on overall structural performance Paper has been re-written, however needs more details: A) basic physical behavior needs more explanation . B) Author did not add  Flow Chart of the entire system, B)  Single, double, and triple pass physical behavior does not describe the effect of  machining is due to strain or residual stress., and does  explain why double pass is best and recommended. C) Data Scatter is not much, and too much emphasis on ANOVA .  Please use the ANOVA statistic and plot Probability of Failure Vs. Response (i.e., strain, or stress) under several  machining conditions (1st pass, 2nd pass, 3rd pass). D) Sensitivity plots are appropriate  Both Conclusion and Abstract must be modified again to substantiate the claim. 

PLEASE ANSWER THESE NOTES QUESTIONS  (BELOW) and SHOW WHERE in TEXT as Yellow color

DETAILS:

Fig 8 show low sutta. Paper improved, too much relying on ANOVA, need more explanation of mechanism of 2nd Pass (Best result) Plane again answer Note 1-6 where is it addoomed in txt.

Most notes are not answered.

Note 1: Introduction (Line 24) is good, however needs to describe the more details such as  relationship between roughness, and residual stress as surface traction (is not explained)

Note 2: Figure 1- Please develop a new table and describe all parameters.

Note 3: I suggest plot of Temperature -time -History (TTH) by measurement of temperature or analysis be plotted as new Figure showing Double Pass Turning compared. 

Note 4 : Figure 3- Need More Detail- Please describe why Double pass has less roughness. What causes it. Explanations are not clear“   ROOT CAUSE PROBLEM MAY BE RESIDUAL STRESS, or RESIDUAL STRAIN.  IF you have Test Data please show the effect of The surface or Hardness, etc.

Note 4 : Figure 4. Shows no difference in Micro hardness between Single, double, and triple .  Why? Please explain. If Residual stress has any effect

Note 5: Figure  5A THIS IS VERY IMPORTANT Figure . why Residual stress is negative in single and third pass.  In axial direction.  Please explain is not clear what caused it. Figure  5B Why Radial stress in double pass is reduced.  This needs to be clearly explained as what parameters are casing this behavior.

Note 6 : Table 4.  Explain the behavior. Of Double pass Crack density. It simply does not make sense.).

Note 7 : Figure 6 EXPAND IT.  As the explanation of double pass 2 . May be in Double pass crack is lower due to grain boundary (Intergranular)  vs. Single and Triple Pass Grain crystallization (Trans granular crack)

Note 8 : Table 6-8.  ANOVA: Describe ANOVA technique in detail, Why Pearson correlation coefficient used

Note 9. Plot all the important parameters causing) Roughness, b) residual stress, and c) Hardness. Plot sensitivity of all parameters in Table 6-9 in the form of CDF (Cumulative Distribution Function) , Probability of Failure (POF)  Vs. Response (Strain, Stress, Etc)  and in more detail Plot
